# The Treatment Alliance and the Provision of Removable Dentures: Exploring the Emotional Work of the Dental Team

**DOI:** 10.3390/dj12110344

**Published:** 2024-10-28

**Authors:** Barry John Gibson, Sarah R. Baker, Tom Broomhead, Bilal El-Dhuwaib, Nicolas Martin, Heba R. Salama, Gerry McKenna, Anousheh Alavi

**Affiliations:** 1School of Clinical Dentistry, University of Sheffield, Sheffield S10 2TN, UK; s.r.baker@sheffield.ac.uk (S.R.B.); t.w.broomhead@sheffield.ac.uk (T.B.); n.martin@sheffield.ac.uk (N.M.); hrsalama1@sheffield.ac.uk (H.R.S.); 2Centre for Public Health, Queen’s University Belfast, Belfast BT9 5AG, UK; g.mckenna@qub.ac.uk; 3Haleon, Weybridge, Surrey KT13 0DE, UK; anousheh.x.alavi@haleon.com

**Keywords:** treatment alliance, dentures, emotions, patient management

## Abstract

**Background:** Research has demonstrated that the loss of one or more natural teeth can be an emotionally traumatic experience that mirrors processes associated with bereavement. There remains scant literature examining the role of emotions in dental encounters. One such exception is the literature on the idea of the ‘treatment alliance’ in dental encounters. The aim of this paper is to explore the role of the ‘treatment alliance’ in dentist–patient encounters. **Methods:** Data were collected from clinical observations, semi-structured interviews and focus groups exploring the experience of tooth loss and how the treatment alliance shaped the patient journey. Data analysis was conducted using the grounded theory method influenced by phenomenology. Coding was conducted using NVIVO and the unit of analysis was the treatment alliance. **Results:** Twenty participants took part in the interviews (eleven male; nine female; 22–86 years; mean age = 58.9 years). Observations were carried out with a further fourteen participants (seven male; seven female; 50–101 years; mean age = 62.2 years). The paper draws on four cases taken from the observational data to illustrate important dynamics underpinning how the treatment alliance varied. These case studies are then used as the basis for a critical discussion of the importance of the treatment alliance in dentistry. **Conclusions:** The treatment alliance acted as an important moderator in the clinical encounter, helping to influence successful outcomes. An important foundational component of the treatment alliance was the degree of emotional work the dental team conducted when ‘getting to know’ unfamiliar patients. Whilst emotions were an important factor in dental encounters, their acknowledgement and management were not essential to all successful outcomes. Much more research is therefore needed into the role of emotional labour in dental encounters.

## 1. Introduction

Tooth loss has been identified as a deeply emotional and personal experience [1,2,3,4] that can mirror the psychological process of bereavement. When patients lost all their teeth, Fiske et al. [3] found that they would go through the same emotional stages associated with loss (denial, anger, depression, bargaining and acceptance). The total loss of teeth can therefore result in widely varying responses, with some struggling to accept this loss, whilst at the same time experiencing significant physical and social impacts on their wellbeing. The combination of these processes often led to an atmosphere of secrecy, isolation and taboo. The process of having a denture fitted to replace missing teeth has also been described as a traumatic experience that has remained poorly understood. In the past, the approach in dentistry was to explain treatment outcomes primarily from the perspective of the personality characteristics of the patient (health locus of control and neuroticism). The evidence for this was at best weak, and at worst possibly ‘upside down’ [3]. Researchers failed to recognize that having a denture fitted was, itself, potentially traumatizing. Instead of attributing outcomes to the personality or characteristics of participants suffering from tooth loss, Fiske et al. [3] identified that the grieving process and disability may well be the key factors that needed to be acknowledged *and managed*. Researchers later demonstrated that the trauma of tooth loss could lead to significant ‘biographical disruption’ [5], which refers to a significant change in someone’s sense of who they are and who they are becoming. All of this suggests that there is a role for emotional support through the treatment process by the dental team. There are very few models of ‘provider–patient encounters’ [6] that acknowledge the centrality of emotions to the loss of teeth.

In a review of the models for understanding ‘provider–patient encounters’ in medicine and dentistry, Sondell and Söderfeldt [7] provided an early overview of normative versus empirical models. Normative models of patient care were developed out of dissatisfaction with the biomedical model. This dissatisfaction led to a biopsychosocial model that stressed shared decision-making as opposed to paternalism and the asymmetric relationship first recognized by Parsons [8,9]. Some of the models mentioned by Sondell and Söderfeldt included the ‘model of participation’, ‘the mutual trust model’, ‘the interactive model’ and ‘the relationship of mutuality’. All these models focus on how involved the patient becomes in their care. Normative models of the dentist–patient relationship [7] contrast with empirical models because the latter seek to accurately describe such encounters.

Empirical models focus on analysing the significance of the provider–patient relationship, and its impact on the overall outcome of encounters. These include locus of control theory [10], the health belief model [11], Ajzen Fishbein’s theory of reasoned action [12] and self-efficacy theory [13]. The focus within these models is on communication, in understanding the psychosocial dynamics between dentists and patients during visits, something that is frequently overlooked. In this literature, it is argued that high-quality technical dentistry does not always associate with high patient satisfaction [14]. In addition, process variables play a role in the outcomes that develop from such encounters [7]. Sondell and Söderfeldt [7] argued that the most important requirement for research into dentist–patient interactions ought to be a focus on a systematic theory of dentist–patient communication. This kind of theory, they argued, should focus on background, process and outcome variables in addition to hypothesizing about how these are related. Yet many of the models in their review do not account for emotions, despite these featuring heavily in discussions about tooth loss.

To our knowledge, the only approaches that explicitly seek to analyse the impact of emotions in oral and dental care is the work of Zhou et al. [15] and Freeman [6]. The work of Zhou et al. [15] seeks to apply the Verona assessment scheme to dentist–children encounters and would best be described as an empirical approach, whereas Freeman’s [6] approach is best described as normative.

### The Treatment Alliance

Freeman [6] described a psychodynamic viewpoint on the dentist–patient relationship (since the time of writing the original piece on the treatment alliance, the dental team has been expanded to include other professional groups such as dental therapists and hygienists. In addition, there is now greater recognition of the central importance of the whole dental team in patient care. In recognition of this, the paper will hereafter discuss the treatment alliance in relation to the whole team as opposed to simply the dentist–patient relationship. This is in recognition of Professor Freeman’s inclusive approach to dental research in general), presenting the encounter as a resource enabling patients to accept care. In this relationship, the patient possessed the ability to accept (use) the work provided by the dental team, and was also able to accept the ‘uniqueness’ of the team. Freeman, always concerned that there was a persistent danger of inequalities in dental encounters, sought to emphasize the importance of ‘the work’ the team provided and the ‘ability’ of patients to accept this work. In these early formulations Freeman [6], following Szasz and Hollender [16], explained that the individual qualities of the dental team and patient were central to the relationship. So much so that ‘the real relationship’, where the dental team and patient engage with each other’s uniqueness, was central to the success of treatment outcomes. The ‘real relationship’ is an ideal relationship. She defined the treatment alliance as a progressive development of this real relationship, whereby the patient’s anxieties and concerns with respect to dental treatment are manifest and the patient is able to recognize the uniqueness of the dentists’ skills or personal qualities, and as such ‘values them’ [6]. The treatment alliance also refers to the ‘ability’ of the patient to recognize that they are part of the narrative associated with their oral care as it develops. This involves recognition that they need to take responsibility, set goals and work with the dental care professional to achieve those goals.

How has the treatment alliance been developed in oral and dental research?

Authors have sought to develop the idea of the treatment alliance in various ways. Woods [17] revisited the idea with reference to psychodynamic theory by explaining, with examples, how dynamic shifts between each person’s psychodynamic wellbeing might affect dental encounters. Additionally, Yuan et al. [18] developed the treatment alliance into the ‘treatment alliance triad’, to explain the relationship between dental healthcare professionals, parents and preschool children. Yuan et al. [19] also proposed a model for ‘person-centred care’ in dentistry based on a development of the treatment alliance. This model focused on Cohen’s [20] proposal that central variables to the process of oral care were dental anxiety, cost, and perceptions of need. In summary, the following elements of the treatment alliance appear to be the most salient in this literature:**Communication**: Effective communication is crucial for understanding the patient’s concerns, explaining treatment options, and ensuring the patient is well informed about their oral health [19].**Trust**: Patients must trust their dentist to provide competent and ethical care. This trust is built through open communication, empathy, and the dental professional’s demonstration of expertise [19,21].**Shared Decision-Making**: In a strong treatment alliance, decisions about dental treatment are often made collaboratively between the dental professional and the patient. This involves discussing treatment options, risks, and benefits so patients can make informed decisions about their oral health [6,19].**Empathy**: Empathy and understanding can help create a more positive treatment alliance. Patients are more likely to feel comfortable and engaged in their care when they believe their dental professional understands their concerns [18].**Respect for Patient Autonomy**: The treatment alliance involves respecting patients’ autonomy, acknowledging their right to make healthcare decisions and respecting patients’ values and preferences [22].**Outcomes**: it has subsequently been claimed that a strong treatment alliance is associated with better patient satisfaction, adherence to treatment plans, and overall health outcomes. It contributes to a positive dental experience for the patient and fosters a sense of partnership in maintaining and improving oral health [6,17,19,20].With the exception of Freeman’s original article [6] which did use empirical case studies, and the work of Yuan [18], there remain very little data exploring the dynamics of the treatment alliance (it was the intention of Professor Freeman and the primary author to revisit these ideas in the light of these data before her untimely death). This is important because without any form of empirical investigation, the theoretical model proposed remains under-explored. The key research question driving the current paper is—what is the role of the ‘treatment alliance’ in dentist-patient encounters?

## 2. Materials and Methods

The data informing this paper were collected in a multi-method qualitative study using narrative interviews, targeted participant observations and a focus group. The research approach combined grounded theory methods informed by phenomenology [23,24]. Full details of the study population and sampling can be found in our previous paper [25]. The research team was composed of clinical dentists, social scientists (psychology, geography and sociology) and those working in the dental industry. The main study involved collecting narrative data from adults across the United Kingdom in addition to data on targeted participant observations collected from a primary care dental setting in northern England and undergraduate clinics in a teaching dental hospital. The clinical settings were deliberately split between experienced practitioners and novice practitioners who were being trained. Ethical approval was granted by the University of Sheffield’s research ethics committee (application 047005) and NHS ethics (application 312943). In this paper, we use pseudonyms for participants’ names. This part of the study aimed to examine the work of the dental team and patient in the clinic to investigate how fitting a denture was achieved. Observations were recorded in a diary. Finally, three months after the targeted participant observations, debriefing interviews were conducted with all patients and dentists. Eight participants were invited to attend a final focus group to share their experiences with each other about their journey. Details of the journey patients undertake are reported elsewhere [25].

### Data Analysis

As stated previously, the data reported here are derived from grounded theory methods coupled with phenomenological analysis [23,24]. Two members of the team (BG, HS) explored the data in detail, meeting to discuss the analysis and then sharing drafts with the wider team. These approaches are combined because they complement each other. Phenomenology enabled researchers to focus on the lived and embodied experience of tooth loss, and how the treatment alliance shaped the patient journey and the relative success of having a denture fitted, while grounded theory was used to conceptualize key stages of the pathway being investigated. Coding was conducted using NVIVO (Version 1.7.1) [26]. In this paper, the unit of analysis is the treatment alliance defined as ‘the degree to which a real relationship is realized throughout the process of care’. Our goal was to draw on this framework as a heuristic device to examine the extent to which we could observe elements of the treatment alliance in the clinical observations as well as the follow-up interviews. Data are presented through observations and information from the debriefing interviews, triangulated with material taken from the narrative interviews where relevant. This reflects our approach, very much informed by grounded theory, to discover ‘what is going on’ in the treatment alliance and how this is reflected in the process of adapting to removable dentures [27].

## 3. Results

In this paper, we triangulate data from our interview participants (Table 1) with data from our observations (Table 2).

This paper examines the dimensions of the treatment alliance as outlined by Freeman [6] and colleagues [18,19]. The analysis seeks to examine how the treatment alliance manifested in the data. This also involves examining potential limiting factors in the impact of the treatment alliance. To provide a detailed account, four participant’s stories will be presented. Each of these are provided because they highlight the complexity of clinical encounters associated with denture provision.

Narrative 1: Geoff

Geoff was a new patient and we first met him when he turned up for his second appointment. Having just recently had teeth extracted from his upper and lower arch, he now required a denture to be fitted. Throughout his appointments, including checking the wax models, Geoff was very quiet, despite the best efforts of Shakil and the Nurse to communicate with him. Shakil took a lot of time to explain everything that was going on, trying to get Geoff to talk about the sensations in his mouth. He coached Geoff by saying, for example, “can I get you to stick your tongue out and move it from side to side please, thank you”. At the try-in stage, he carefully explained, “we are going to try them in and if everything is okay we will get the proper denture made up next week. We can only make minor changes after today so it is very important to get them as comfortable as possible. It is important to also be aware that the denture will change how you bite so it will feel a bit different”. Not only was he coaching Geoff, he was also responding to the uniqueness of the denture as it fitted within Geoff’s rapidly changing dentition. This involved making sure it fitted the gap left by the extractions as well as matching the shade to his existing teeth.

“I think it’s a C3 shade. No, let’s check C4. C4 matches the bottom teeth.”

“What do you think?”

“It’s all right.”

“As you can see, the canines are always darker.”

“As long as it’s not a dazzling Hollywood smile that looks ridiculous I’m okay.”

Shakil also talked about the future problems Geoff was going to be faced with.

“Your teeth may need quite a bit more work and we might find that this denture will eventually need to be changed. Some of your teeth are in quite bad condition. They will need looking after. This is what a transitional denture means.”

It was only when the actual dentures were being fitted that Geoff started to become more animated. It was obvious that Geoff’s mouth was very sensitive. The presence of his denture, the thing he was going to have to wear when he left the clinic, was animating the whole interaction. Both he and Shakil were working on the denture. Geoff realised that this was something he was going to have to live with and so was actively taking a role in helping get it shaped so that it could sit within his mouth comfortably and firmly.

“Again, is it okay?”

“It’s all right.”

“How’s the bite?”

They worked to understand the process of putting the denture in and taking it out. This involved a little bit of negotiation with Shakil, who would hold a mirror up for Geoff so that he could look at where he was grabbing the denture with his fingers. It didn’t take very long for Geoff to gradually understand the motion of the denture within his mouth. He was able to remove it, click it in, and then take it back out again. It was only really a matter of a few attempts before the whole thing was being done fluently.

Throughout this process, they would talk to each other. At one point, Geoff stated “now I don’t know how to do it.” Such interactions indicated that they were both training Geoff to develop a new skill to live with his denture. Throughout this process, Shakil coached Geoff, getting him to try different positions to exert pressure on the denture whilst they sought to remove it.

“Yeah that’s tight, so we’re going to have to adjust it again”—they took the denture out several times, made tentative adjustments, and gave the denture back to Geoff who would try it in with his fingers and then gradually learn how to remove the denture from his mouth. It was obvious that they were developing both know-how and bodily skills, and the ability to place the denture and take it out. At the final appointment, it was clear that Geoff understood the challenge he was being confronted with. Shakil talked about the transitional nature of this denture and explained to Geoff that it was quite a big step to move from a situation where you had nothing in your mouth to having a full denture. He encouraged him to keep his natural teeth for as long as possible and to only add to the denture where it was absolutely necessary. It was quite surprising how animated Geoff had become now that he had his denture. We all looked at his reflection in the mirror and he looked so much younger, his face was a better shape and he was smiling.

A fundamental aspect of the interaction was that both sides of the discourse were preserved. While both the dental team and the patient had different expectations, their unique perspectives remained. A completely shared understanding was impossible; after all, Geoff had to wear the denture and Shakil had to help make it fit. Geoff was confronting the fact that his oral health was changing and that having a denture was something he had to live with. Shakil, on the other hand, understood oral health, and knew Geoff faced some difficult times ahead because he had experience with other patients in similar situations. He was aware that Geoff may find a full denture difficult to adjust to. These different perspectives persisted even at the end of the interaction. Their uniqueness persisted. This is what Freeman [6] and Szasz and Hollender [16] were referring to when they argued that differences are to be tolerated and worked with in the interaction.

At the follow-up interview three months later, it emerged that Geoff was no longer using his denture. He had left the clinic and bought a sandwich, and when it really hurt to bite into it he took his dentures out and never put them back in. This was despite the advice he was given in the clinic. He was coached to just wear his dentures for a short time initially, and then very gradually work from wearing them to eating with them and speaking with them. This was because, as Shakil explained to many of his patients, his mouth would initially try to force his dentures out and would need time to adjust. Geoff had a long experience of problems with his teeth and every time he went to a dentist more work was needed. He said:


*“..I didn’t used to make a six-month appointment. Because usually there was always something breaking on my teeth because one, I grind my teeth in my sleep and that. Yeah, there was usually another damaged tooth or something before an appointment I’d made, so I just used to go as and when I was needing something fixed. So, it was generally a couple of times a year just for that alone anyway.”*

*(Geoff, 50, Male, Unemployed)*


His principal problem was the functionality of his denture and an emerging concern that the denture was going to cause more damage to his teeth.


*“For me, when I had them fitted and everyone at the dentist was saying, “Oh, yeah, you’ll look younger,” and this, that and the other, I thought, well, I’m not really (Laughs) bothered about any of that. It’s just whether I can eat this or crunch on that and it’d be better than struggling with all these teeth missing and what have you and gaps in between them and stuff.*



*But yeah, having had the denture in for that day, when I took it out I’d got a chip on one of the teeth at the side. Either side of where my front teeth were, I’ve got the two top teeth. Both of them have got little bits of chips in them now. On one of them it happened, I think, taking that denture in and out. It caused a chip that day. I thought, well, that’s not ideal.*



*I think when I do go to the dentist next for getting these fillings sorted out I’ll try to see if they can do anything to patch that up because otherwise that might end up causing more problems later. I’ll be trying to get it repaired if I can.”*

*(Geoff, 50, Male, Unemployed)*


Geoff contrasted dramatically with Jenny.

Narrative 2: Jenny

Jenny had a long experience with her dentures and had been attending this dental clinic for over 20 years, seeing many different dental team members. She was very chatty and comfortable in the clinic and already had a longstanding relationship with Shakil. At her first appointment, she talked in a “very businesslike fashion” about the fact that her current denture was moving and needed to be replaced. She had a cobalt chrome denture but she was going to have an acrylic denture fitted. The reason for fitting the acrylic denture was because the teeth to which the cobalt chrome denture was fitted were no longer stable enough for that denture to remain. She indicated throughout that she had “had this done several times before” and that she was “okay with it”. Shakil was very careful to continue to explain each stage, but it was obvious that Jenny knew exactly what to expect from the treatment. At times, the nurse had commented that “you might find there are bits around your mouth” and Jenny had replied that she had “found that before.” Indeed, as her appointments finished she explained that she would “not see any other dentist” and had “stormed out of seeing that other dentist.” Jenny was therefore a very experienced patient (in contrast to Geoff) and had a very strong relationship with Shakil. She was so comfortable that at the try-in stage, she indicated that she wanted the new denture to be more visible, something that Shakil was keen to help her with. Throughout treatment, she and Shakil actively discussed the denture, its colour, and fit. Her primary goal was to make sure her teeth were visible and ‘comfortable’—this related to the denture being secure in her mouth and “not slipping”. Shakil fitted the denture and adjusted it for her. He checked this and Jenny said:

“I can feel it here.” She was pointing to the bottom right of her mouth.

“It’s all right, we will make some adjustments today,” before making the adjustments and checking these with Jenny.

“Can you take it out? Can you put it in and out?”

Shakil also explained “…to take it easy, start with soft food and build it up, don’t go straight to hard foods, try different things but gently increase the pressure on the denture and don’t rush.”

“We will give you a leaflet on how to adapt to your new denture.”

“Oh brilliant, you can see them, brilliant! I like the adjustment! … There is a big difference, I can see them at the top and I can feel it.” Jenny was delighted with this new denture and was very excited to be wearing it finally. She had a great big grin flashing her new top teeth at us all. She said that:

“Most of it is that no one knows I have a denture because you don’t want people knowing, do you?”

This was a hugely successful case. Shakil knew her very well and this made a huge difference in terms of knowing what her expectations were. This was nearly a full denture and so it was very difficult to make and adapt to. Jenny had very few teeth left. A few missing teeth and a stable bite is relatively easy to measure and communicate to the technician. In Jenny’s case, there were very few teeth left in the top of her mouth. Adding to this problem was how much her teeth were showing because they were quite high in her mouth, and they needed to adjust this for her. An important factor in this case from Shakil’s point of view was that Jenny is a very experienced patient who was moving from a partial denture towards a full denture. The change was, relatively speaking, easy. Yet Shakil was not aware of the intimate nature of the denture; he did not know, for example, that she had stopped going out and was avoiding intimacy with her partner because her old denture was so loose.

Narrative 3: Rob

Rob had requested his first denture because his tooth loss had become so bad that it was now interfering with his ability to get work, to see his children and enjoy his everyday life. He had stopped going out and was restricted in his ability to socialise. He was extremely self-conscious of his oral condition. Vee, his student dentist, explained this to me at the first appointment. Rob had lost a lot of teeth, he had several missing teeth at the front of his dentition. Worse still, he was facing losing the rest of his teeth. This denture was going to require further additions at a later date.


*“The worst thing about it is the gaps in your mouth, it gets you down. I get depressed. I think, am I able to get through this? I’m going to say that I know this is my own fault—I smoked for 40 years. Don’t get me wrong, it is very much my own fault. I stopped because I had to and then I needed the dentures.”*

*(Rob, 56, Male, Unemployed)*


Throughout treatment, Rob struggled with a very strong gag reflex, choking constantly and finding it very hard to go through treatment. Rob was also deeply embarrassed about the state of his dentition and struggled to communicate at times with the dental team. Vee, as a student dentist, was extremely careful with Rob, constantly encouraging him, paying attention to his discomfort and taking a lot of time to ensure he was comfortable. Taking the first and second impressions was very difficult, not just because of Rob’s gag reflex, but also because his current dentition was so mobile that the process was painful and deeply uncomfortable. Vee took a lot of time to talk him through the impressions and coached him really carefully. At one point he had to go through two sets of impressions in the same session because the impression was incomplete. The tray was refitted back into Rob’s mouth and we could see that he was getting agitated. He struggled to tolerate the process of fitting the tray in and out due to his gag reflex, indicating that, “it’s been in there for too long, too long, that’s all.”

Vee was extremely careful with Rob, coaching him throughout, explaining that they were almost done and praising him for his tolerance. At the next appointment, Rob appeared at the clinic and informed Vee that he had lost his previous bridge tooth as a result of the disturbance from the impressions. This resulted in a complication that Vee had not anticipated (she later explained this was her first denture!) and would require an additional tooth for the denture. Rob’s periodontal status had become extremely unstable, but he had withdrawn from the treatment he was being provided for this elsewhere in the hospital. He also reported that as a result of taking previous impressions, his teeth were now very painful—“I can’t talk, I can’t do anything,” he said. This emphasised the need for Vee and the clinic to take a record of his withdrawal from treatment elsewhere, but also to get Rob’s denture fitted as quickly as possible. At one point the tutor was forced to step in to reassure Rob. Rob was sitting in the chair clearly very fed up with the clerical staff at the clinic.

“It’s been a nightmare. I called in and they got angry with me and told me to stop calling. My priority is with these loose ones. I can’t go on with them and they are really painful too.”

The tutor came over and started to talk to Vee and Rob. “Tell me about this gentleman.”

“When the impressions were taken, the next morning I lost this one you had fixed and I couldn’t talk because of the vibrations. I can’t go on.” He was distraught. The tutor reassured Rob and talked him through the problems he was having with the unstable teeth. The tutor advised him that they could be removed and fitted to the denture as “additions”.

“Well we can add these to the denture, they would be called additions, and we can remove your teeth and then add new ones to the denture as we go.”

“Oh, that would be awesome. How close am I to getting the denture fitted?”

At this point the tutor looked to Vee who said that the “try-in would be in two weeks.” They eventually extracted more teeth, added these to the denture and finally got to the fit stage. Throughout this process, Rob’s physical difficulties with the impression trays and even the wax models continued to be challenging. Vee continued to carefully coach him, praising his patience. During the final try-in Rob almost retched several times, struggling to keep the denture in for longer than 30 s. Both Vee and the tutor sat with him and explained that his mouth would reject the prothesis, just as Shakil had done with Geoff in the primary care clinic. He eventually left the clinic. At the debriefing interview three months later, I found that Rob was doing well. He explained the significant challenge of having to train his body over a prolonged period of time to get used to the denture. He was absolutely determined to have his teeth fixed.


*“It’s depressing, you go from looking half-decent to looking like a bit of a hobo, if you know what I mean. You see yourself as not looking the same as you did before.”*


Then later in the interview:


*“You had a difficult time as well with the impressions and that, didn’t you?”*



*“Yeah, my gag reflexes. Better now obviously because I’ve got them in. I don’t think the tutor thought I’d be able to do it. … It was horrible. First few days I couldn’t put them in at first, then I couldn’t keep them in for more than a couple of minutes. Like I said, I needed to look better, I needed myself to look better than I was, so that’s why I persisted. It’s not easy now but it’s better, it’s a lot better than it was before.”*

*(Rob, 56, Male, Unemployed)*


Rob’s outcome was good, but clearly he was deeply anxious because his dentition remained unstable. He was anticipating that he would need more work in the near future. Nonetheless, he had managed to wear his denture and from this perspective he was doing as well as he could expect.

Narrative 4: Lisa

Lisa was a retired schoolteacher who had a long relationship with her dentist Eddie. They were fitting a lower denture which was going to be based on a cobalt-chrome framework. This was a complex denture to make because Lisa had lost quite a few teeth and so there were a lot of factors affecting how it sat in the spaces between her existing teeth. Lisa was experienced as a patient, this was not her first denture. The challenge for Eddie was that they were attempting to extend her bite backwards to replace some of her missing back teeth. This was not going to be easy. Throughout the process, Eddie coached Lisa but she was quite familiar with the treatment process and had little difficulty with the impressions. The whole atmosphere in the clinic was one of a relaxed familiarity between Eddie and Lisa.

When they were discussing Lisa’s dentition, Eddie talked about what mattered to her the most, including the two gaps at the front of her lower dental arch and filling in the gaps at the back. Eddie knew she wouldn’t mind the bigger denture, and whilst it might be more to get used to, she had plenty of previous experience. We had a discussion about the complexities of working with dentures that are modified over time. Lisa would need to have clasps on her denture that would be fitted to the back teeth. Metal dentures are much more complex to work with and adding new structure to them can be very difficult. On top of this, when the dentist is making the denture fit there has to be an estimate of the stability that can be tolerated by the patient and at the same time what kind of gap might be allowed in order to get the denture in and out. The impressions were complex and required an extra visit, but this did not bother Lisa who easily accommodated them. The denture was quite tight when it was fitted initially and Lisa had some difficulties removing it. The metal framework also made the denture difficult to adjust causing some problems by sticking into her gums here and there. For example, as they adjusted the denture Lisa remarked “awwwww, it’s scratching me there.” Throughout the process, Eddie would work with Lisa to adjust the denture as the important balance between the tightness of fit and flexibility proved hard to achieve. But they both persevered and eventually got it fitting so it could be removed and refitted without causing a lot of pain.

“I use my tongue there to help remove it”—there were several further adjustments to the denture and this involved trying it in, taking it out and taking little bits off here and there. Each time Eddie would ask Lisa to take the denture out she would struggle to do so. This then involved gradual adjustments to make it looser so that it could be removed and placed back in the mouth with ease. There was a working compromise between goodness of fit and comfort and this is what they were both working on in the clinic. Clearly Lisa had different levels of tolerability, and the denture had to be forced to yield, but not so much that it became too loose. At the follow-up interview, Lisa was happy with the denture although it had taken more than five visits for it to settle in. She was happy with Eddie and the care she received, always felt included in her treatment and felt that Eddie had listened to her needs and responded to them. The denture, on the other hand, had continued to resist every attempt to make it a perfect fit. She said that she had developed a habit of not eating her tea with the denture in because her previous denture had been so uncomfortable.


*“I’m wearing it now. I could eat an apple if I fancied one. I could carry on doing exactly what I want to do, until teatime, and then take it out. (Laughter)”*



*“So, you wouldn’t eat your tea with it in, then?”*



*“No, and sometimes… I mean, last night we had… I burnt my finger last night on it. Look, I’ve got a blister.”*



*“Bless you.”*



*“We had cottage pie and vegetables, and so I could have practically put that in my mouth and nearly swallowed it without chewing it, but I still took it out and put it there. I think it’s, maybe, habit, which we’ve not mentioned yet. It might just be habit, because I ate lunch with it in, and breakfast. No, I think I had… I usually clean my teeth after breakfast and put it in then, because I’d have to—might have to—do it again, (Laughs) so it’s to save doing it twice.”*

*(Lisa, 72, Female, Retired Teacher)*


Each of the cases can be summarized according to the key variables in the treatment alliance (communication, trust, provider empathy and the real relationship) along with the conditions and outcomes of care (see Table 3). The conditions that most affected each case were whether or not the patient had prior experience with dentures, a prior relationship with the dental team, and the complexity of the denture. Outcomes were rated on the basis that the participant could use the denture with minimal disruption to their daily life. Seven patients were defined as having a ‘solution achieved’, indicating outcomes that involved some sort of compromise acknowledged by the patient and the dentist and which involved ongoing daily adjustments to the denture. This could involve using adhesives, or modifying how the denture is worn during the day, for example, removing it in the evenings or to eat meals at home.

Two participants were no longer using their denture. One had stopped using his denture because he was only going to be without his front tooth for a short time and “it was more hassle than it was worth anyway”. This participant was paying for private dentistry and would soon be having a bridge provided. He was therefore in no real need of the denture and so did not need to adapt to it either. He turned it into a joke amongst his friends, calling himself a pirate. Geoff’s case is very revealing because he was new to dentures, had two made that were both complex, and his ongoing oral health was very unstable. He had low trust in dentistry as a whole and did not really trust the dental team he was seeing either. This was despite the team doing everything they could to show empathy. In the end, Geoff could not develop any form of relationship. The treatment alliance in this case was weak and so its absence certainly predicted a negative outcome for Geoff. It is not surprising that he never returned to the clinic for further support even when it was suggested that he should do so.

## 4. Discussion

This is the first qualitative study looking at the treatment alliance in clinical dental encounters. It combines multiple qualitative methods of data collection (observations, interviews and focus groups) and is therefore innovative and rigorous. The findings of this research suggest that the treatment alliance is best understood as an *important moderator of outcomes in clinical encounters*. It is important to note that here, we are talking about the psychosocial processes associated with tooth loss and the provision of dentures. We have shown that, whilst tooth loss can be an emotionally significant life event [1,2,3], a dental team that can build a good working relationship with patients can have a positive impact and enable patients to overcome the challenges associated with tooth loss. This can lead to better adaptation to living with dentures. These findings are overwhelmingly positive and, whilst ongoing impacts might remain (seven out of fourteen participants in the observational study had significant ongoing impacts on everyday life), many patients were nonetheless comfortable with their treatment outcomes.

The findings highlight that there needs to be a discussion about the role of empathy in the treatment alliance. Empathy involves being able to understand the other’s perspective, recognize emotional distress and show compassion. When the fitting of the denture was difficult to achieve, an empathetic response from the team served to enable patients to build trust, improve communication and enable the ‘real relationship’ [6]. This suggests that the Verona assessment system [15,28] has an important role to play in helping to analyse how emotions are managed in the clinic.

This research also suggests that improving the skills of dental teams in recognizing their involvement in ‘emotional work’ might enable the development of a stronger therapeutic alliance [28]. More work is needed, however, to explore not only the importance of empathy, but also what the role of emotional management might look like for dental teams. One framework that might be considered for further investigation is the cognitive–emotional decision making (CEDM) framework [29]. It is important to note, however, that whilst the recognition and management of emotions can be important, there are quite a few other variables at play during the encounter. This includes the experience the patient has with dentures, with the team providing care, and high levels of system trust. Participants’ prior experience with dentures and the system of dentistry were important predictors of better outcomes. These variables interacted with the complexity of the denture itself, in support of Sondell et al.’s [7] review. This finding extends previous work exploring the significance of oral care as a life course project [26,30]. Patients with an ongoing embodied knowledge of ‘having work done’ were much more enabled to engage in more work. They could even go beyond poor working relationships with dentists to find a new dentist and get the best outcomes for themselves. This suggests the hypothesis that low interpersonal trust can be moderated by high system trust [31], ‘know-how’ and empathy [26].

Our findings show that empathy was *foundational* to the development of good verbal communication, helping to establish trust and eventually the ‘real relationship’ [6]. This foundational role of emotional management is by no means universal. Whilst the study highlights the significance of the emotional dimension to care, it does not provide in itself a systematic assessment of the role and position of this dimension. It was clear from each of the case studies that the team engaged in ‘emotional labour’. Here, ‘emotional labour’ refers to demonstrating care for the patient’s comfort and experience of treatment. Although this factor is not often discussed in the literature [32], it is nonetheless an important form of ‘social capital’ [33] not widely recognized in dentistry. This was the case for instances when clinical and interpersonal outcomes were excellent, and it was also important when some sort of compromise was going to be needed on the part of the patient.

In some instances, no matter how positive the relationship between the team and the patient, or indeed how experienced the patient was themselves, the solution achieved could fall short of being totally successful. A surprising finding of this study is that these cases were not marked by ‘failure’ but that a relatively successful outcome could be realized when the patient accepted a compromise. A strong treatment alliance enabled such compromises, resulting in very low levels of complaint, and indeed relatively high levels of satisfaction. An important characteristic of the relationship between many of the patients in this study and their dental team was one of gratefulness and warmth, even when the treatment outcomes were less than perfect.

It is also important to recognize that positive outcomes could be achieved in the absence of the treatment alliance. Such instances were relatively infrequent ‘exceptions that proved the rule’. This is an important finding, suggesting the need to avoid ‘essentializing’ the treatment alliance beyond what it can achieve and realizing that there are likely to be limits to the influence of this feature of clinical encounters. More work is needed to explore the boundaries and implications of this important concept.

In conclusion we found that, whilst *a strong treatment alliance on its own was not a necessary nor sufficient cause to produce good outcomes*, it was an important *moderating factor*. It is important that the dental team recognizes the potential importance of emotional labour in clinical encounters, especially in supporting patients through difficult treatment. This can be achieved by demonstrating empathy and engaging with the patient as they undertake their difficult journey through tooth loss.

### Limitations

Although this study combines multiple methods of qualitative data collection (observations, interviews and focus groups) and so should be considered rigorous, it is nonetheless subject to a number of limitations. First, caution should be exercised when applying the findings to a wider population given the size and convenient nature of the sample. Another important limitation is that recruitment for the clinical encounters could often only happen after the initial consultation had been completed. This was to ensure ethical practice in securing ‘consent to contact’ from participants. The study does not always include data from these initial consultations and so has not provided a systematic window into the initial decision-making processes. Finally, the presence of the research team definitely had an impact on the overall process. In some cases, patients who were finding the adjustment to their denture difficult went back to the dentist for further adjustments; this often happened because they were interacting with the research team. This had a significant impact on outcomes by adding additional moderation to the outcomes experienced in such cases. In these instances, some patients listened to the interviewer (BJG) and then went back to have adjustments made to their denture. Not all participants availed themselves of this advice, however. Some participants just did not make that follow-up appointment. It is for this reason that we suggest much more systematic work is needed into the role of emotional labour in dental encounters. Such impacts could have been reduced by conducting the debriefing interviews six months after the final appointment.

## Figures and Tables

**Table 1 dentistry-12-00344-t001:** Participants from the narrative interviews.

Participant Number and Name	Age	Sex	Occupation
Participant 1—Keith	46	Male	Warehouse Manager
Participant 2—Betty	44	Female	Disability Support Advisor
Participant 3—Penny	61	Female	Nurse (retired)
Participant 4—Bill	52	Male	Car Mechanic
Participant 5—Emma	46	Female	Care Home Assistant
Participant 6—Florence	24	Female	University Student
Participant 7—Rose	26	Female	Nurse
Participant 8—Lyana	22	Female	University Student
Participant 9—Venus	71	Female	School Teacher (retired)
Participant 10—Shay	64	Female	Manager (retired)
Participant 11—Ben	75	Male	Gardener (retired)
Participant 12—Paul	64	Male	College Lecturer (retired)
Participant 13—Alex	86	Male	Metal Caster (retired)
Participant 14—Carrie	71	Female	Receptionist (retired)
Participant 15—Michael	71	Male	Mental Health Nurse (retired)
Participant 16—James	73	Male	Youth Worker (retired)
Participant 17—Bobby	66	Male	Engineer
Participant 18—Mark	69	Male	Engineer (retired)
Participant 19—Stephen	75	Male	Teacher (retired)
Participant 20—Roger	71	Male	Builder

**Table 2 dentistry-12-00344-t002:** Summary of participants and outcomes for observations.

Participant Number and Name	Age	Sex	Occupation
Participant 21 Jenny	55	Female	Registered Disabled
Participant 22 Jim	55	Male	Building Manager
Participant 23 Karen	64	Female	Childminder (retired)
Participant 24 Lisa	72	Female	Teacher (retired)
Participant 25 Alicia	101	Female	Housewife
Participant 26 Danny	75	Male	Council Worker
Participant 27 Geoff	50	Male	Unemployed
Participant 28 Tareq	60	Male	Landlord
Participant 29 Fami	64	Female	Disabled
Participant 30 Annette	68	Female	Teacher (retired)
Participant 31 Harry	83	Male	Builder
Participant 32 Kenneth	62	Male	Disabled
Participant 33 Rob	56	Male	Courier, Carer
Participant 34 Jane	68	Female	Factory Operative (retired)

**Table 3 dentistry-12-00344-t003:** The treatment alliance, conditions, process and consequences.

	Conditions			The Treatment Alliance				Consequences	
Participant Number and Name	Experience with Dentures	Prior Relationship with the Dental Team	Complexity	Communication	Trust	Provider Empathy	The Real Relationship (Autonomy)	Return Visits	Outcomes ^1^
Participant 21 Jenny	Yes	Yes	Yes	High	High	High	High	4	V Successful
Participant 22 Jim	No	No	No	High	Medium	High	High	3	Solution achieved
Participant 23 Karen	Yes	Yes	Yes	High	High	High	High	3	Solution achieved
Participant 24 Lisa	Yes	Yes	Yes	High	High	High	High	5	Solution achieved
Participant 25 Alicia	Yes	No	Yes	Low	Medium	High	High	2	Solution achieved
Participant 26 Danny	Yes	No	Yes	Medium	Low	High	High	0	V Successful
Participant 27 Geoff	No	No	Yes	Low	Low	High	Low	0	Not using
Participant 28 Tareq	No	Yes	No	High	High	High	High	0	Not using
Participant 29 Fami	Yes	Yes	Yes	High	High	High	High	2	Solution achieved
Participant 30 Annette	Yes	No	No	High	High	High	High	0	Solution achieved
Participant 31 Harry	Yes	Yes	Yes	High	High	High	High	1	V Successful
Participant 32 Kenneth	No	No	Yes	High	High	Medium	High	1	V Successful
Participant 33 Rob	No	No	Yes	Low	Low	Medium	Low	0	V Successful
Participant 34 Jane	Yes	No	Yes	Medium	Medium	Medium	Medium	1	Solution achieved

^1^ This indicates an outcome that involved some sort of compromise that was acknowledged by *both* the patient *and* the dentist. Such outcomes involve ongoing daily adjustments to the denture.

## Data Availability

The original contributions presented in the study are included in the article, further inquiries can be directed to the corresponding author.

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
