# Peer review of "The Treatment Alliance and the Provision of Removable Dentures: Exploring the Emotional Work of the Dental Team"

_dentistry, 2024, doi:10.3390/dj12110344_

Round 1
Reviewer 1 Report
Comments and Suggestions for Authors
The article titled "The Treatment Alliance and the Emotional Work of the Dental Team" presents an important and innovative study on the role of the treatment alliance in dental encounters, focusing on the emotional dynamics between dental teams and patients. This study is particularly significant given the emotional distress associated with tooth loss and denture fitting, areas often underexplored in dental literature.
The research is commendable for its rigorous methodological approach, combining grounded theory with phenomenological analysis, and employing multiple qualitative methods, including narrative interviews, participant observations, and focus groups. This multi-method approach allows for a nuanced understanding of the treatment alliance, making the study robust and insightful.
One of the paper's strengths lies in its clear and thorough articulation of the concept of the treatment alliance, as it integrates both historical and contemporary perspectives, particularly drawing on Freeman's psychodynamic framework. The inclusion of detailed patient narratives is another strong point, as it humanizes the data and provides concrete examples of how the treatment alliance plays out in real-world settings.
However, the article has several weaknesses that need to be addressed before publication. First, the paper suffers from repetition, particularly in the introduction and literature review sections. The same points about the emotional impact of tooth loss and the importance of the treatment alliance are reiterated multiple times, which could be streamlined to enhance readability and maintain the reader's engagement.
Moreover, the discussion, while comprehensive, occasionally strays into tangential topics that dilute the focus on the treatment alliance. For instance, the mention of other psychological models and theories, although relevant, sometimes feels overextended and distracts from the central theme. A more focused discussion would strengthen the paper’s core argument.
The paper's conclusion, while generally strong, could benefit from a more critical reflection on the limitations of the study. The authors do mention the impact of the research team's presence on the participants, but this section is relatively brief. A deeper discussion on how this might have influenced the findings, as well as other potential biases, would provide a more balanced and transparent account of the research process.
In terms of language, the paper is generally clear, but there are occasional lapses into jargon or overly complex sentences that may obscure the intended meaning. Simplifying some of these expressions would make the paper more accessible, particularly to a wider audience that may not be familiar with all the specialized terminology.
Finally, the references, while extensive, show some inconsistencies in formatting that need to be corrected to meet academic standards. Ensuring uniformity in citation style throughout the paper is crucial for its professional presentation.
In conclusion, while this paper presents a valuable contribution to the field of dentistry, particularly in understanding the emotional dimensions of dental care, it requires some revisions to address issues of repetition, focus, and clarity. Additionally, a more critical engagement with the study's limitations and a thorough review of the references will help to enhance its scholarly rigor before publication.
Author Response
Thank you for your helpful comments on this paper. We have attempted to include all of your suggestions and feel the paper is strengthened as a result.
Reviewer comment: First, the paper suffers from repetition, particularly in the introduction and literature review sections. The same points about the emotional impact of tooth loss and the importance of the treatment alliance are reiterated multiple times, which could be streamlined to enhance readability and maintain the reader's engagement.
We have revised the introduction to remove repetition and streamline the material.
Reviewer comment: "Moreover, the discussion, while comprehensive, occasionally strays into tangential topics that dilute the focus on the treatment alliance. For instance, the mention of other psychological models and theories, although relevant, sometimes feels overextended and distracts from the central theme. A more focused discussion would strengthen the paper’s core argument."
We agree that the discussion was a bit too convoluted, and so we have reduced it significantly. This meant collapsing paragraphs and removing repetition.
Revier Comment: "The paper's conclusion, while generally strong, could benefit from a more critical reflection on the limitations of the study. The authors do mention the impact of the research team's presence on the participants, but this section is relatively brief. A deeper discussion on how this might have influenced the findings, as well as other potential biases, would provide a more balanced and transparent account of the research process."
We have added more information about the impact of the researcher's presence and reflected on the complexity of these effects in the conclusion.
Reviewer comments: "In terms of language, the paper is generally clear, but there are occasional lapses into jargon or overly complex sentences that may obscure the intended meaning. Simplifying some of these expressions would make the paper more accessible, particularly to a wider audience that may not be familiar with all the specialized terminology."
Thank you. The main author is visually impaired. The whole team has reviewed the manuscript, and we use Grammarly as part of our assistive technology. Much of the complexity was from overloading the paper with too much detail and not paying attention to streamlining the discussion. We welcome this opportunity to respond to these very helpful and detailed comments and do feel the paper is much stronger as a result.
Reviewer Comment:
"Finally, the references, while extensive, show some inconsistencies in formatting that need to be corrected to meet academic standards. Ensuring uniformity in citation style throughout the paper is crucial for its professional presentation."
Here, we use reference management software and always seek to clarify our referencing in the final draft. These have now been checked and fixed.
Reviewer 2 Report
Comments and Suggestions for Authors
Dear Authors,
thank you very much for this interesting manuscript focusing on an important part of dentistry. The manuscript is well written.
Some suggestions might help.
Title: I recommend using a more detailed and sharpened title representing the aim and outcome of your work when dealing with prosthetic patients. This might increase reader´s interest.
Abstract: Please give more information about the methods, the included patients and the outcome of your study. I would prefer to give a clear aim.
Introduction: the introduction is fine. At the end I would prefer to give a clear hypothesis and null hypothesis of your study. As it is a part of the whole study I suggest including the specific characteristics of the investigated patient group.
Material and Methods: Was the study registered in any clinical trial registry? Please give information. why did you include 34 patients? Any sample size calculation?
Results: Number, Name Age and Occupation of the participants were included in the manuscript. Why did you include the names? I suggest removing the names of the patients throughout the manuscript.
Conclusion: Please give a clear conclusion for clinical dentistry at the end referring to your results. This might help reader´s to include your findings in their daily experience.
Please add the required parts at the end of the manuscript including Author Contributions, Funding, Institutional Review Board Statement, Informed Consent Statement, Data Availability Statement, Conflicts of Interest.
Author Response
Thank you for your kind comments.
Reviewer's comments:
"Title: I recommend using a more detailed and sharpened title representing the aim and outcome of your work when dealing with prosthetic patients. This might increase reader´s interest."
We have adjusted the title to be more specific and it now mentions removeable dentures.
Reviewer comment: "Abstract: Please give more information about the methods, the included patients and the outcome of your study. I would prefer to give a clear aim."
We do state the aim of the study in the abstract. We give a clear indication of the methods stating clearly 3 methods of data collection and we show what form of data analysis was used. We are not sure what else would need to go into the abstract. We have removed one sentence to try and make the outcome clearer.
Reviewer comment: "Introduction: the introduction is fine. At the end I would prefer to give a clear hypothesis and null hypothesis of your study. As it is a part of the whole study I suggest including the specific characteristics of the investigated patient group."
As this is a qualitative study, a null hypothesis is inappropriate. We do appreciate your comment about the patient group. But we also recognise the paper is very long already and so refer the reader to another paper where the details of the group being studied are carefully cited.
Reviewer comment: "Material and Methods: Was the study registered in any clinical trial registry? Please give information. why did you include 34 patients? Any sample size calculation?
Thank you for this comment. This is not a clinical trial, and we do not test hypotheses, so calculating the sample size is not appropriate. The discussion of number 34 is in the paper we refer to in the results. However, we definitely did not make this clear in our original submission—we have added a line to explain to readers that further details of the sample can be found in our previous paper.
Reviewer comment: "Results: Number, Name Age and Occupation of the participants were included in the manuscript. Why did you include the names? I suggest removing the names of the patients throughout the manuscript."
Thank you for spotting this - we use 'pseudonyms' but had not stated this in the paper. We are sorry for this.
Reviewer Commnet: "Conclusion: Please give a clear conclusion for clinical dentistry at the end referring to your results. This might help reader´s to include your findings in their daily experience."
Thank you. We have added the following statement:
"It is therefore important that the dental team recognizes the potential importance of emotional labor in clinical encounters. Especially in supporting patients through difficult treatment. This can be achieved by demonstrating empathy and engaging with the patient as they undertake their difficult journey through tooth loss."
Reviewer Comment: Please add the required parts at the end of the manuscript including Author Contributions, Funding, Institutional Review Board Statement, Informed Consent Statement, Data Availability Statement, Conflicts of Interest."
These details have now been appended to the paper. Thank you"
Reviewer 3 Report
Comments and Suggestions for Authors
The manuscript explores the concept of the "treatment alliance" in dentist-patient interactions, particularly in the context of tooth loss, and highlights its significance in achieving successful clinical outcomes. The inclusion of case studies offers practical examples that illustrate the dynamics of the treatment alliance, making the findings more relatable and applicable to real-world dental practice. By utilizing a combination of clinical observations, semi-structured interviews, and focus groups, the study delves into how emotions and the treatment alliance shape the patient journey in dental care.
Overall, the manuscript owns its impact and provide clearer guidance for both research and clinical practice in dentistry. I would like to support this manuscript.
Author Response
Thank you for taking the time to review this manuscript.